# Profiling Malnutrition Prevalence among Australian Rural In-Patients Using a Retrospective Census of Electronic Medical Files over a 12-Month Period

**DOI:** 10.3390/ijerph17165909

**Published:** 2020-08-14

**Authors:** Laura Alston, Megan Green, Vincent L Versace, Kristy A. Bolton, Kay Widdicombe, Alison Buccheri, Didir Imran, Steven Allender, Liliana Orellana, Melanie Nichols

**Affiliations:** 1Global Obesity Centre (GLOBE), Institute for Health Transformation, Faculty of Health, Deakin University, Geelong, VIC 3220, Australia; steven.allender@deakin.edu.au (S.A.); melanie.nichols@deakin.edu.au (M.N.); 2Deakin Rural Health, Faculty of Health, Deakin University, Geelong 3220, Australia; vincent.versace@deakin.edu.au; 3Colac Area Health, Colac, VIC 3250, Australia; Mgreen@cah.vic.gov.au (M.G.); KWiddicombe@cah.vic.gov.au (K.W.); ABuccheri@cah.vic.gov.au (A.B.); dimran@cah.vic.gov.au (D.I.); 4Institute for Physical Activity and Nutrition, Deakin University, Geelong 3220, Victoria, Australia; kristy.bolton@deakin.edu.au; 5Biostatistics Unit, Faculty of Health, Deakin University, Geelong, VIC 3220, Australia; l.orellana@deakin.edu.au

**Keywords:** malnutrition, in-patients, rural, malnutrition risk, census, electronic medical files

## Abstract

In-patient malnutrition leads to poor outcomes and mortality, and it is largely uninvestigated in non-urban populations. This study sought to: (1) retrospectively estimate the prevalence of malnutrition as diagnosed by dietetics in the rural Australian setting; (2) establish the proportion of all patients at “nutritional risk”; and (3) explore associations between demographic and clinical factors with malnutrition diagnosis and nutritional risk. A retrospective census was undertaken of medical files of all patients aged ≥18 years admitted to a rural hospital setting over a 12-month period. Logistic regression was used to explore associations between malnutrition diagnosis, nutritional risk and patient-related factors. In total, 711 admissions were screened during the 12-month period comprising 567 patients. Among the 125 patients seen by dietitians, 70.4% were diagnosed with malnutrition. Across the total sample, 77.0% had high levels of nutrition related symptoms warranting a need for further assessment by dietitians. Malnutrition diagnosis by dietitians was associated with being over the age of 65 years, and patients had higher odds of being admitted to a residential aged care facility following discharge. In this rural sample, the diagnosis rate of malnutrition appeared to be high, indicating that rural in-patients may be at a high risk of malnutrition. There was also a high proportion of patients who had documentation in their files that indicated they may have benefited from dietetic assessment and intervention, beyond current resourcing.

## 1. Introduction

Malnutrition is defined as a physical condition resulting from lack of absorption or intake of nutrition leading to altered body cell mass and composition, which contributes to poorer health status, diminished physical and mental function and impairs clinical outcomes when present with disease [1]. Malnutrition among in-patients is a global challenge, with several international studies estimating that the prevalence of malnutrition in hospitalized patients in developed countries is 20–60% [2,3,4,5,6,7,8]. The majority (60–80% of admitted patients, especially elderly) are at risk of malnutrition and are likely to develop malnutrition when unwell and admitted to an acute hospital ward [2,9]. Malnutrition among in-patients leads to a high preventable burden on the healthcare system [2,5,6] and is associated with poorer health outcomes and quality of life, longer length of stay, extended recovery times, higher likelihood of readmission and higher mortality [2,5].

It is recommended that health services implement routine malnutrition screening practices to identify those at risk of malnutrition who may require further nutrition assessment and intervention [10]. Some jurisdictions including parts of Denmark, the Netherlands, the United States and the United Kingdom have mandated routine screening of in-patients with this being achieved dependent on hospital accreditation [5,11]. Appropriate and timely screening can be completed by nursing or medical staff who do not have nutrition-specific training, and it has been found to decrease the rate of hospital-acquired malnutrition [3]. Implementation of routine mandatory screening in the health service setting has been documented to be poor, with a study in the Netherlands finding that only 8% of admitted patients received mandatory screening without any errors (such as underestimated nutrition status or incomplete documentation) [2].

The diagnosis of malnutrition in the in-patient setting is also difficult, with no widely accepted “gold standard”. It is recommended that patients are first screened for malnutrition risk, and then those who are judged to be at increased risk, should be seen by a dietitian for further assessment and diagnosis [1]. In 2019, the Global Leadership Initiative on Malnutrition consensus criteria for the diagnosis of malnutrition included that a clinician (usually a dietitian) must do the assessment and the diagnosis should be based on the presence of at least one phenotypic criterion (e.g., change in weight) and one etiologic criterion (e.g., reduced food intake) [5].

In Australia, rural populations experience poorer health status relative to metropolitan areas [12,13]. Malnutrition prevalence may follow a similar pattern to other health conditions, with a higher burden in rural areas [14]. However, this is largely unknown as the prevalence among rural patients is under-investigated, with no evidence available on the burden of malnutrition risk among in-patients admitted to health services located in places classified as inner and outer regional areas of Australia by the Australian Statistical Geography Standard (ASGS) [15]. Rural in-patients have different demographic profiles (including age and education levels) and different healthcare access and resources [12,16]. These factors may influence malnutrition prevalence in rural areas, and subsequently the needs of rural health services in addressing nutritional related issues in their service communities, including the need for mandatory screening in these settings.

To contribute to the understanding of these factors, we undertook the first retrospective census to characterize malnutrition prevalence and nutritional risk in regional in-patients in Victoria, Australia. Nutrition risk, in this context, was indicated when patients had enough nutrition related documentation on indicators of malnutrition in their files which would warrant further assessment by dietitians. Other retrospective file audits have used dietitians to extract data on malnutrition risk and make clinical judgements on the level of nutrition risk (patients likely requiring further assessment) or malnutrition within an in-patient population [3,17,18,19]. For example, Larsen et al. used nutrition related chart notes in medical files (such as reduced oral intake) to retrospectively flag patients that may have benefited from nutrition intervention but were not identified during their admission to understand gaps in malnutrition identification and treatment [18]. Bohringer and Brown used the Malnutrition Screening Tool scoring guide to retrospectively score medical files, based on documentation of nutrition symptoms, to determine if oncology patients were appropriately referred to dietetics in a rural clinic setting [19]. In this study, we sought to understand the prevalence of diagnosed malnutrition, along with understanding what proportion of patients may have benefited from further nutrition assessment due to documentation in clinical notes.

The aims of this study were to conduct a census of all in-patients admitted to a regional hospital in Victoria, Australia, in the period 2017–2018 to:estimate the prevalence of malnutrition as diagnosed by dietetics;establish the proportion of all patients at “nutritional risk” (those who may have needed dietetic assessment/intervention) by retrospectively scoring electronic medical files; andexplore associations between demographic and clinical factors with malnutrition diagnosis and nutritional risk.

## 2. Materials and Methods

This study design included a retrospective census of electronic medical files that was undertaken by clinical dietitians to collect demographic, clinical and nutritional data, guided by existing malnutrition screening tools.

### 2.1. Ethics Approval

This project was approved by the Deakin University Human Research ethics committee and a letter of confirmation was received from the Barwon Health ethics committee (2019-058). A waiver of consent was approved for the retrospective census of electronic medical files. Access to the data was granted to the researchers only.

### 2.2. Sampling and Data Extraction

The study was conducted at a regional hospital in Victoria. A list of all admissions from July 2017 to June 2018 was obtained from the hospital’s health information department in 2019. Admissions were eligible if the patient was at least 18 years old, not pregnant, with a length of stay of three or more days. Admissions of less than three days were excluded due to the nature of documentation in electronic medical files and the likelihood that there would not adequate information to score the files according to the documentation of nutrition symptoms as dietitians would be less likely to have seen patients at less than three days discharge, due to resourcing, time taken to receive referrals and there being no dietitians available on weekends in this context.

If patients had more than one admission during the census period, the longest admission was included in the analysis. In the sample, patients were general admissions to the acute ward to capture a representative sample, as no major surgeries are conducted at the health service. Data on admission reasons were highly heterogeneous in documentation, which precluded analysis of these factors. Two experienced clinical dietitians with malnutrition expertise undertook the data collection. One clinical dietitian completed the majority of the data extraction and cross-checked findings and file scoring with the second clinical dietitian when needed (10% of files), with 100% agreement.

Information collected from each admission included: age at admission (in years), length of stay (LOS) in days, whether the patient lived in the larger rural township or smaller surrounding rural communities, whether the patient was seen by a dietitian during the admission, whether the patient was diagnosed with malnutrition by a dietitian, nutrition related symptoms (as guided by Patient Generated Subjective Global Assessment (PG-SGA)) and discharge destination. Previous studies have used data from files on “nutrition related notes” to retrospectively understand if patients should have been flagged as needing nutrition intervention [17,18,19] with data extraction conducted by experienced clinical dietitians, as guided by malnutrition screening tools. Data from the medical files were assumed to be complete for each admission, as required by clinical documentation guidelines at the health service.

PG-SGA [20] is a malnutrition screening tool which assigns patients a score to identify those needing nutrition intervention and guides the subsequent triage of nutrition intervention (Table 1). This tool, along with the clinical judgement of the experienced dietitians, was used to guide the data extraction of nutrition symptoms from the electronic medical files and to allocate risk scores, based on PG-SGA scoring guidelines. The tool has been validated at the patient bedside to have high reliability and is simple enough to be used by health professionals other than dietitians to triage referrals and outlines a scoring system for nutrition symptoms. For example, the tool assigns a point system for each nutrition symptom, e.g., 3 points for loss of appetite, 2 points for nausea or no points for no change in food intake [20]. The tool has a high sensitivity for identifying malnutrition risk, similar to the validated Subjective Global Assessment (SGA) [21]. The tool was used to guide the collection of nutritional indicator information from the medical files including information regarding level of oral intake, recent weight loss, age, conditions, fever, steroid medications, nutrition impact symptoms (poor appetite, nausea, constipation, mouth sores, no taste/poor taste, swallowing difficulties, pain, vomiting, diarrhea, dry mouth, sensitivity to smells and early satiety) and levels of activities and function (for access to the tool, please visit http://pt-global.org/). “Nutritional risk”, as guided by the literature [18,20,21], was defined as having sufficient nutrition symptoms within files to retrospectively warrant further dietetic assessment and potential intervention. Scores up to 3 were categorized as “low nutritional risk” and above 3 as “high nutritional risk”, indicating a need for dietetic assessment and potential to prevent/address malnutrition based on the PG-SGA triage guidelines [20]. Table 1 shows the categories of PG-SGA scores, triage recommendation and nutrition risk level used in this study. Permission was sought to use the PG-SGA tool for this study from the copyright owner Dr Faith Ottery.

The catchment region of this hospital included two different levels of remoteness as classified by standard geography measures. Patients address was used to classify the patient as living either within the main township or surrounding small rural towns. The main township where the hospital is located is a “Medium Rural Town” (MM4) and the outer rural areas are defined as “Smaller Rural Towns” (MM5) according to the Modified Monash Model (MMM) [22]. The MMM is increasingly used in Australia to classify areas into different levels of remoteness. The MMM was developed based on the Australian Statistical Geography Standard and uses populations and road distances to classify areas into remoteness categories [15,22].

### 2.3. Statistical Analysis

All statistical analyses were conducted using Stata SE Version 15 (StataCorp LLC, College Station, TX, USA) [23]. A *p*-value < 0.05 was considered statistically significant. *T*-test for continuous variables and chi-squared tests for proportions were used to compare patient characteristics between remoteness, i.e., those living in medium or small rural towns (MM4 and MM5 Modified Monash categories, respectively) [22]. Logistic regression was used to assess the association of demographic and clinical characteristics with: (1) the diagnosis of malnutrition among those seen by a dietitian; or (2) those at nutritional risk among the whole sample. A final model adjusting for all the factors simultaneously was also fitted. The model included the following dichotomized variables based on previous malnutrition literature [3,17,24,25]:Age (under or over the age of 65 years at admission)Length of stay (3–7 days, or more than 7 days)The remoteness of the patient’s current residence (in the “Medium Rural Town” or surrounding “Small Rural Towns”)Sex (male/female)

Discharge destination (as four categories: (1) returned to place of current residence; (2) transferred to another health service; (3) a new admission to a Residential Aged Care Facility (RACF); or (4) deceased).

## 3. Results

In total, 567 patients were screened by clinical dietitians and retrospectively scored (totaling 711 admissions). There was a higher proportion of females in the sample (60.5%). Patients were aged between 19 and 102 years, with the mean age of 70.6 years. The mean length of stay for the total sample was 6.3 days (median of four days), with a range of 3–74 days. The majority of patients currently resided within the main township of the health service catchment (75.0%) and were discharged home or to their residential aged care facility (72.6%). Patients residing in the Medium Rural Town were significantly older than the patients residing from the outer rural areas (*p* < 0.05). A lower proportion of patients residing in the Small Rural Towns were over the age of 65, compared to those residing in the main township, at the time of their admission (60.6% compared to 72.0%, *p* ≤ 0.05). Patients residing in the Small Rural Towns also had a lower mean length of stay (LOS) of 5.7 days compared to 6.6 days for those who resided in the township.

Among all patients who were eligible over the 12-month period, 77.0% were assessed to be at “high nutritional risk” (and would have been considered to require dietetic assessment and potential intervention) when retrospectively screened. Of the total sample, 125 patients (22% of all patients) were seen by a dietitian and of these 88 were assessed to be malnourished (70.4% of the patients seen by dietitians). A higher proportion (73.7%) of patients residing in the Medium Rural Town were diagnosed with malnutrition compared to than those living in the Small Rural Towns (59.3%), but the difference was not statistically significant. The scoring of files showed that similar proportions of patients were at a high nutritional risk in both the Medium Rural Town (77.8%) and Small Rural Towns (74.7%).

Table 2 shows demographic and clinical characteristics, nutritional risk and malnutrition diagnosis by remoteness in the sample of patients assessed by a dietitian (*n* = 125). The mean age was higher than the overall sample (75.7 years) and these patients had a longer length of stay (10.6 days on average). A large majority of patients seen by the dietitian (117 patients, 93.6%) had a high nutritional risk score in the retrospective screening. Eighty-eight patients (70.4% of those seen) were diagnosed with malnutrition by the dietitian.

Patient’s characteristics, according to whether they were diagnosed with malnutrition by the dietitian are shown in Table 3. The age of patients diagnosed with malnutrition was significantly higher on average (79.3 years) than those who were not diagnosed (67.0 years) (*p* ≤ 0.001). Patients diagnosed with malnutrition also had a significantly longer length of stay of 11.6 days compared to a mean of 7.8 days in patients not diagnosed with malnutrition (*p* = 0.03). Of this sample, 17.1% of the patients diagnosed with malnutrition resided outside of the township in smaller rural towns. Most of the patients (37.5%) diagnosed with malnutrition were either discharged home or back to their RACF, and 36.4% were discharged and admitted to a RACF for the first time. Of the patients diagnosed with malnutrition by a dietitian, 87 of these had a high nutritional risk score according to their electronic medical files.

The odds ratios of malnutrition diagnosis were estimated under univariate and multivariate logistic models. Models included age dichotomized as under/over the age of 65, LOS dichotomized as over/under seven days admission, rurality of residence (MM4 or MM5) and place of discharge. In both the univariate and the adjusted models, being over the age of 65 years was associated with higher odds (*p* = 0.03) of being diagnosed with malnutrition and patients diagnosed with malnutrition were more likely to be a new admission to RACF. There were no significant associations by gender and remoteness (Table 4).

Table 5 shows the univariate and multivariate logistic regression models for odds of having a high nutritional risk score according to remoteness and demographic characteristics in the overall sample. In the univariate models, length of stay was significantly associated with having a high nutritional risk score, with having an admission of seven days or fewer being strongly associated with a lower score (OR: 0.16, 95% CI, 0.07, 0.37). However, in the multivariate models, all associations disappeared except for an increased likelihood of having a high score if patients had a length of stay of more than seven days. Patients who were given a high nutritional risk score had significantly higher odds of being seen by a dietitian (OR 5.6, 95% CI 2.6, 11.8) (data not shown in table).

## 4. Discussion

This study is the first retrospective census of electronic medical files to describe the prevalence of malnutrition of Australian rural in-patients and the proportion of patients at high nutritional risk. To the best of our searching, we could not identify any other studies that include a complete census of electronic medical files in a rural sample and no census studies in metropolitan settings. This may be due to the recent uptake of electronic medical files in health service settings, as paper-based census studies would be highly resource intensive and impractical. This study found a malnutrition diagnosis rate of 70.4% among patients assessed by dietitians. Although direct comparison cannot be made, due to different sampling and screening methods, this is higher than predictions in Australian metropolitan studies. It is plausible that malnutrition in rural areas could be higher than in metropolitan areas, given these populations tend to experience higher rates of chronic disease, reduced socioeconomic status, poorer health status and reduced access to healthcare and healthy foods compared to their metropolitan counterparts [12].

It is difficult to determine the amount of data in other Australian studies that may have come from rural patients admitted to metropolitan hospitals; however, the results were not stratified by place of residence by the authors. For example, the prevalence of malnutrition diagnosed by dietetics across 56 hospitals in both Australia and New Zealand was found to be 32.0%; however, the results were not stratified by patient’s place of residence prior to admission [26]. Across two tertiary hospitals, using the Subjective Global Assessment in a random sample of metropolitan based in-patients (*n* = 819), 36.0% were diagnosed with malnutrition [27]. The estimate from this study is also higher than one prospective screening study (*n* = 608 patients) conducted in the Northern Territory (rural Australia) that found a malnutrition rate of 41.7% (95% CI 40.1%, 52.3%), using screening by the subjective global assessment tool [28]. Another consideration in interpreting these results is that we included patients with an admission of three days or more, due to the level of information in the electronic files, which may have led to a sicker or more elderly sample than if all admissions were included. In these shorter admissions, it would be expected that the patient would have been less likely to be seen by a dietitian, particularly if the admissions were over a weekend when allied health staff are not available.

This is the first study to use clinical judgement and the PG-SGA tool to guide a retrospective analysis of nutrition symptom information in medical files and showed that 77% of the total sample may have benefited from a dietetics assessment during their admission, due to being at high nutritional risk. This risk level is impossible for the current level of dietetic resourcing to address in this rural context. Dietetics were only able to assess 22% of patients in this sample, over 12 months. Even if this low proportion were due to a lack of referrals or patients declining assessment, an increase of referrals to 50% of patients to get closer to the estimated risk level of 77% would still be unlikely to be feasible. However, it does also indicate a need for better or mandatory screening of patients on admission and justifies involving additional staff, such as nurses, to implement simple nutrition strategies (such as initiating the ordering of additional snacks for patients with nutrition symptoms) when dietitians are not available. Barker et al. recommended a need for increased resourcing for dietetics in Australia in order to address the risk of malnutrition among in-patients and improve identification and treatment, and these data support the notion that this recommendation would especially apply to rural areas [5]. A consideration for policy may be that the implementation of mandatory screening practices needs to prioritize high-risk populations, such as those in rural areas. The level of nutritional risk could also be under-estimated, because any symptoms that were not documented (e.g., change in weight or reduced food intake) were assumed to not be present during the admission, however due to limitations in documentation this may have not been precisely true. Based on this estimate of risk and the current level of resourcing for dietitians and allied health staff generally in rural areas [29,30], current services would not be able to address this level of nutrition risk in reality. More screening studies in rural samples would be needed to determine the true level of resourcing required to address malnutrition in comparable rural areas in Australia.

This study found, consistent with previous literature, that patients diagnosed with malnutrition had a significantly longer length of admission and were older when compared to those who were not diagnosed with malnutrition [5]. Being over the age of 65 years was most strongly associated with a malnutrition diagnosis by dietetics in this rural sample, consistent with other research on the link between ageing and malnutrition risk [1,25]. This was also true for length of stay, as patients who were diagnosed with malnutrition by the dietitian stayed a mean of almost four days longer than their non-malnourished counterparts. This is similar to findings in an Australian metropolitan study that found malnourished patients had a significantly longer length of stay of around five more days, and malnourished patients were around eight years older than non-malnourished patients [27]. Given the high proportion of patients identified at risk, but not seen by a dietitian, future research into predictors of referrals to dietitians is needed in the rural context.

In our study, a higher proportion of residents living in the medium rural town were diagnosed with malnutrition compared to those who resided in the small rural towns, although this was not significant. This may have been due to the “migration effect”, whereby people living in more remote areas will move into main townships (closer to health services) when they start to experience age-related health issues [31]. As Gregory (2009) explained, this movement could lead to under-estimation of the health burden in more rural and remote areas. In this study, it is possible that many of the patients currently residing in RACF had originally come from outer rural areas which was not captured in this study [31].

There are multiple strengths to this study in that we completed a census of all admissions over a 12-month period that met our inclusion criteria, leading to a representative sample in this context. The data were extracted by two clinical dietitians with knowledge of malnutrition, with the use of the PG-SGA to guide data extraction on nutritional risk—this confers an accurate estimation of risk in the sample. Study limitations include the nature of retrospective audits of medical files, such as the inability to check the accuracy of notes with the patient or health professional perspective. It is possible that some clinicians did not document all symptoms present during the admission, and, for research purposes, it was assumed that each medical record was a complete record of the admission. It does however provide a detailed picture that would not be able to be collected prospectively without time and staff required to undertake screening of every patient admitted. In addition, patients who were seen by dietitians may be subject to selection bias, which would be expected in the clinical setting. A potential limitation is that one dietitian completed the majority of the screening of the files, which was cross-checked by a second dietitian in the event of uncertainty. However, to reduce errors, the data extraction was guided by the validated PG-SGA and is therefore likely to be highly accurate, along with the clinical judgement of both dietitians. Patient weight was not always documented in this sample, which meant that scoring may have been under-estimated. Data on admission reasons and diagnoses were highly heterogeneous in the documentation by medical staff, which precluded analysis of these factors and is a limitation of the retrospective audit design.

## 5. Conclusions

This census of electronic medical files of rural in-patients found a diagnosis rate of malnutrition of 70.4% by dietitians among those who were assessed and estimated around 77% may have benefited from dietetic assessment or intervention in the full in-patient sample. The remoteness of patient’s residence was not associated with malnutrition diagnosis or risk, however being over the age of 65 years was strongly associated with malnutrition, consistent with global research on malnutrition among in-patients. Further studies are needed to understand the prevalence of malnutrition and risk among rural Australians admitted as in-patients to adequately resource health services and reduce the burden on both patients and the healthcare system.

## Figures and Tables

**Table 1 ijerph-17-05909-t001:** Nutrition triage recommendations based on the PG-SGA score.

Score	PG-SGA Guidelines	Nutrition Risk (Need for Further Assessment/Intervention by Dietetics)
0–1	No nutrition intervention required at this time. Re-assessment on routine and regular bases	Low
2–3	Patient education potentially needed but no nutrition intervention	Low
4–8	Requires intervention by dietitian to assess malnutrition in conjunction with nurse/physician as indicated by scored symptoms	High
>9	Indicates a critical need for nutrition intervention	High

Notes: Adapted from Ottery, 2001 ©.

**Table 2 ijerph-17-05909-t002:** Demographics and screening characteristics of all patients assessed by a dietitian by remoteness.

Patient Characteristics	Patients Residing within Medium Rural Township (MM4) (*n* = 99)	Patients from Small Rural Towns (MM5) (*n* = 26)	Total Patients Seen by Dietitians (*n* = 125)	Total Overall Sample (*n* = 567)
Age in years, mean (range)	76.6 (27–97)	72.4 (29–95)	75.7 (27–97)	70.6 (19–102)
LOS, mean (range)	10.6 (3–60)	10.4 (3–74)	10.6 (3.0–74.0)	6.3 (3–74)
Males, *n* (%)	33 (33.3)	14 (53.9)	47 (37.6)	224 (39.5)
Females, *n* (%)	66 (66.7)	12 (46.2)	78 (62.4)	343 (60.5)
High nutrition risk scores, *n* (%)	91 (91.9)	26 (100)	117 (93.6)	437 (77.1)

Notes: LOS, length of stay; “MM4”, Modified Monash category of “Medium rural towns”; “MM5”, Modified Monash category “Small rural towns”.

**Table 3 ijerph-17-05909-t003:** Characteristics and comparison of patients seen by the dietitian by diagnosis of malnutrition.

Patient Characteristics	Not Diagnosed with Malnutrition *n* (%)	Diagnosed with Malnutrition *n* (%)	Total Sample (% Total Sample Seen by Dietitians)
Seen by dietitian	37 (29.6)	88 (70.4)	125 (100)
Mean age (range)	67.0 (29–97)	79.3 (27–96) *	75.7 (27–97)
LOS (range)	7.8 (3–21)	11.6 (3–74) *	10.5 (3–74)
Resides MM4	26 (70.3)	73 (82.9)	99 (79.2)
Resides MM5	11(29.7)	15(17.1)	26 (20.8)
**Sex**			
Females	22(59.5)	56 (63.6)	78 (62.4)
Males	15 (40.5)	32 (36.4)	47 (37.6)
**Place of discharge**			
Back to RACF/home	27 (73.0)	33 (37.5)	60 (48.0)
Deceased during admission	1 (2.7)	3 (3.4)	4 (3.2)
Transferred to other health service	6 (16.2)	20 (22.7)	26 (20.8)
New admission to RACF	3 (8.1)	32 (36.4) *	35 (28.0)

Notes: LOS, length of stay; RACF, Residential Aged Care Facility; “MM4”, Modified Monash category of “Medium rural towns”; “MM5”, Modified Monash category “Small rural towns”; * denotes statistically significant difference (defined as *p* ≤ 0.05).

**Table 4 ijerph-17-05909-t004:** Odds of malnutrition diagnosis by dietitians according to demographic factors (univariate and multivariate models).

Patient Characteristics	Odds Ratio (95% CI) Univariate Model	*p*-Value	Odds Ratio (95% CI) Multivariate Model	*p*-Value
**Age**				
Under the age of 65 years (ref)	1.0		1.0	
Over the age of 65 years	4.32 (1.73, 10.80)	*p* = 0.002	3.11 (1.13, 8.53)	*p* = 0.03
**LOS**				
More than 7 days (ref)	1.0		1.0	
7 days or less	0.37 (0.17, 0.82)	*p* = 0.014	0.63 (0.23, 1.70)	*p* = NS
**Resides**				
MM4 (ref)	1.0		1.0	
MM5	1.93 (0.79, 4.70)	*p* = NS	2.10 (0.76, 5.90)	*p* = NS
**Sex**				
Females (ref)	1.0		1.0	
Males	0.86 (0.39, 1.89)	*p* = NS	1.22 (0.50, 2.99)	*p* = NS
**Place of discharge**				
Back to RACF or home (ref)	1.0		1.0	
Transferred to other hospital	2.65 (0.93, 7.51)	*p* = NS	2.01 (0.62, 6.55)	*p* = NS
New admission to RACF	8.50 (2.34, 30.67)	*p* = 0.01	5.25 (1.30, 21.80)	*p* = 0.02
Deceased	2.38 (0.23, 24.22)	*p* = NS	1.03 (0.86, 12.30)	*p* = NS

Notes: 95% CI, 95% confidence interval; “MM4”, Modified Monash category of “Medium Rural Towns”; “MM5”, Modified Monash category “Small rural towns”.

**Table 5 ijerph-17-05909-t005:** Demographic factors and the association with a high nutrition risk score (univariate and multivariate models).

Patient Characteristics	Odds Ratio (95% CI) Univariate Model	*p*-Value	Odds Ratio (95% CI) Multivariate Model	*p*-Value
**Age**				
Under the age of 65 years (ref)	1.0		1.0	
Over the age of 65 years	1.14 (0.75, 1.74)	NS	0.90 (0.57, 1.40)	NS
**LOS**				
More than 7 days (ref)	1.0		1.0	
7 days or less	5.98 (2.71, 13.20)	*p* < 0.001	4.70 (2.04, 10.82)	*p* < 0.001
**Resides**				
MM4 (ref)	1.0		1.0	
MM5	1.19 (0.77, 1.86)	NS	1.06 (0.66–1.67)	NS
**Sex**				
Females (ref)	1.0		1.0	
Males	0.73 (0.49, 1.08)	NS	0.73 (0.58, 1.11)	NS
**Place of discharge**				
Back to RACF or home (ref)	1.0		1.0	
Transferred to other hospital	2.17(1.14, 4.17)	*p* = 0.02	1.76 (0.89, 3.45)	NS
New admission to RACF	3.2 (1.35, 7.74)	*p* = 0.008	1.68 (0.65, 4.32)	NS

Notes: 95% CI, 95% Confidence Interval; “MM4”, Modified Monash category of “Medium rural towns”; “MM5”, Modified Monash category “Small rural towns”. All patients who died during their admission had a high nutrition risk score and were dropped from the regression model.

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
