# Peer review of "Profiling Malnutrition Prevalence among Australian Rural In-Patients Using a Retrospective Census of Electronic Medical Files over a 12-Month Period"

_ijerph, 2020, doi:10.3390/ijerph17165909_

Round 1

Reviewer 1 Report

The study was designed and performed correctly. The chosen methods are adequate to the aim of the study.

However, the results and conclusions only confirm the findings and conclusions from already existing literature (Baker et al. 2011, Gout et al. 2009, Lazarus et al. 2005) - the suggestion of the need of compulsory nutritional risk screening in Australia while it is already compulsory in some other countries. 

According to well-established problem in the international standpoint (the need of identification of undernutrition in patients), the article can rather highlight the already known problem in a direct reference to this particular country population. That is why I suggest to include the information about this local (Australia) impact in the title of the paper and in the abstract. 

The Authors mention that similar study could not be performed prospectively due to resources requirements (line 304). What kind of the resources are meant? PG-SGA that was applied in this study is a low-cost tool and can be performed any other member of the medical staff (physician, nurse), not exactly by a dietician (http://pt-global.org/).

The Authors should describe in details how the symptoms were transformed into PG-SGA score (lines 127-137).

Do authors know any reasons why nutritional status assessment is still facultative in Australia while mandatory in other countries? (line 46)

There is a reference missing in line 293.

There are minor interpunction corrections needed (double space marks, commas etc.).

Author Response

Please see attachment and thank you for your comprehensive review

Reviewer 2 Report

Alston et al. present a retrospective study of the prevalence of malnutrition in a rural Australian in-patient setting. They set out to identify prevalence of malnutrition and nutritional risk in this population, and furthermore to explore associations between demographic and clinical factors with malnutrition diagnosis and nutritional risk. Malnutrition is an often underapreciated clinical problem, and real life data from a rural setting are of high interest to clinicians working in the same setting, who often lack data from cohorts representing their patient collective.

The paper is well written and has several strengths, namely

(1) Screening of 100% of patients in a 12 month period, minimizing inclusion bias

(2) appropriate cohort size for meaningful statistical analysis.

(3) relevant clinical topic

(4) study representing an understudied Patient collective.

Unfortunately, there are some weaknesses which greatly diminish the enthusiasm for this paper

(1) Missing data points. As usual in a retrospective cohort, there are missing data points as not all data points are initially documented. The authors do not report in how many cases there were missing data points, so that the quality of data is hard to judge. At least at subcohort analysis with the patients for which all data points are available should be performed in order to see wether there is a significant difference between the whole cohort and the subcohort.

(2) The authors report that 77% of subjects were at nutritional risk, but only around 20% were seen by dieticians. This strongly suggests a selection bias in the patients seen by dieticians, and questions the applicability of results found in this subcohort on a bigger patient collective. predictors for referral to a dietician should be investigated and discussed, as they are relevant to the interpreatation of the data presented

(3) Unfortunately, the cohort is not very well defined. There is no data presented on diagnosis or Underlying conditions, as well no Differentiation between surgical and non surgical patients or elevtice and non-elective stays. The results the authors present are very hard to interpret without these pieces of informations, and the reason why those are not included is not clear to this reviewer.

(4) 90% of dataextraction was done by a single investigator, which is a very error prone approach.

Overall the authos investigated an interesting topic, but should better define their cohort in order to substantially improve this paper before publication.

Side note/spell check:

lines 179/180: 'Among all patients who had eligible patients over the 12 month perio ..' seems to be a mistake, as this sentence makes no sense as is.

line 79 Larsen et al, should read Larsen et al.

Author Response

(The authors gave the same response as above.)

Reviewer 3 Report

This study by Alston et alia aims at quantifying the proportion of patients admitted in hospitals and suffering from malnutrition, with a focus on rural areas of Australia. They also correlate these results to demographical characteristics.

From my point of view, the manuscript is well written and easy to understand. However, a few points will need further explanations. These are detailed below, concerning both main and minor points.

Main points :

  1. In the method section (ethics section), authors should explicit who could access the database they present within this study.

  1. Authors should discuss what are the causes of malnutrition in rural/remote Australia. Although speculative, such an information could be paralleled to previously published studies. Besides, since the age of all patients included in this study varied widely (27-97 and 19-102 years old), could authors further discuss the origin of malnutrition ? My understanding would be that older patients could present altered metabolism (pathophysiology) while younger patients could consume unbalanced diets. Should this information be accessible from the database, authors should briefly discuss/mention such results.

  1. In the discussion, authors compare their data (70.4% of malnutrition within the sample seen by dietitians) to the data within Australia as a whole country (32%). Is it possible that the latter includes patients from rural areas that were admitted in metropolitan hospitals ? This is also mentioned on page 7 (lines 249-251) concerning the Northern Territory.

Minor points and typos :

Introduction :

Page 1, lines 40-41, typographical error with references 2, 5 and 11.

Page2, line 55, a coma is missing after “In 2019”.

Page 2, line 57, double spacing in “diagnosis should”.

Page 2, line 58, the full stop is before the reference.

Methods :

Page 3, line 121, the reference should be placed before the coma.

Page 3, line 125, “simply” was written instead of “simple”.

Page 4, line 150 and 153, presence of double spacing.

Results :

Page 5, line 178. I do not understand the meaning of “Among all patients who had eligible patients over the 12 month period […]”.

Page 5, line 182, “dietetics” or “dietitians” ?

Page 5, lines 187-188. Is the sentence missing “in” ? (“in the sample of patients assessed by…”).

Page 6, line 217, I do not understand the sentence “were more likely to be transferred be a new admission to RACF”.

Page 6, line 227, presence of double spacing.

Discussion :

Page7, lines 243-245 lack of reference(s).

Page 7, line 249, presence of double spacing.

Page 7, lines 261-263, I do not understand this sentence.

Page 8, line 275, presence of double spacing.

Page 8, line 281, absence of spacing between “malnutrition” and reference number 5.

Page 8, line 291, “This may have been due the migration effect”. Due to the migration effect ?

Author Response

(The authors gave the same response as above.)

Round 2

Reviewer 1 Report

Last corrections are satisfactory, except for lines 134-135. Authors describe there the way of transforming data into points in PG-SGA. They mention giving "no points for change in food intake", while the change of it ("less than usual") should be given 1 point according to PG-SGA. That part should be revised.

I also suggest including the applied PG-SGA form into supplementary data.

The interpunction also should be checked once again. 
